


# Net community production in the northwestern Mediterranean Sea from glider and buoy measurements

Michael P. Hemming[1], Jan Kaiser[2], Jacqueline Boutin[3], Liliane Merlivat[3], Karen J. Heywood[2], Dorothee C. E. Bakker[2], Gareth A. Lee[2], Marcos Cobas García[2], David Antoine[4,5], and Kiminori Shitashima[6]

[1]Coastal and Regional Oceanography Lab, Centre for Marine Science and Innovation, UNSW Sydney, Sydney, Australia
[2]Centre for Ocean and Atmospheric Sciences, School of Environmental Sciences, University of East Anglia, Norwich, United Kingdom
[3]Laboratoire d'Océanographie et du Climat, Sorbonne Université, Paris, France
[4]Remote Sensing and Satellite Research Group, School of Earth and Planetary Sciences, Curtin University, Perth, Australia
[5]Laboratoire d'Océanographie de Villefranche, Villefranche-sur-Mer, France
[6]Tokyo University of Marine Science and Technology, Tokyo, Japan

**Correspondence:** Michael Hemming (m.hemming@unsw.edu.au)

**Abstract.** The Mediterranean Sea comprises just $0.8\,\%$ of the global oceanic surface, yet considering its size, it is regarded as a disproportionately large sink for anthropogenic carbon due to its physical and biogeochemical characteristics. An underwater glider mission was carried out in March–April 2016 close to the BOUSSOLE and DyFAMed time series moorings in the northwestern Mediterranean Sea. The glider deployment served as a test of a prototype ion-sensitive field-effect transistor

pH sensor. Dissolved oxygen ($O_2$) concentrations and optical backscatter were also observed by the glider and increased between 19 March and 1 April, along with pH. These changes indicated the start of a phytoplankton spring bloom, following a period of intense mixing. Concurrent measurements of $CO_2$ fugacity and $O_2$ concentrations at the BOUSSOLE mooring buoy showed fluctuations, in qualitative agreement with the pattern of glider measurements. Mean net community production rates ($N$) were estimated from glider and buoy measurements of dissolved $O_2$ and inorganic carbon (DIC) concentrations.

Glider and buoy DIC concentrations were derived from a salinity-based total alkalinity parameterisation, glider pH, and buoy $CO_2$ fugacity. The spatial coverage of glider data allowed calculating advective $O_2$ and DIC fluxes. Mean $N$ estimates for the euphotic zone between 10 March and 3 April were $(-17\pm35)$ for glider $O_2$, $(44\pm94)$ for glider DIC, $(17\pm35)$ for buoy $O_2$ and $(49\pm86)\,\mathrm{mmol\,m^{-2}\,d^{-1}}$ for buoy DIC, all indicating net metabolic balance over these 25 days. However, these 25 days were actually split into a period of net DIC increase and $O_2$ decrease between 10 and 19 March and a period of net DIC

decrease and $O_2$ increase between 19 March and 3 April. The latter period is interpreted as the onset of the spring bloom. The regression coefficients between $O_2$ and DIC-based $N$ estimates were $0.25\pm0.08$ for the glider data and $0.54\pm0.06$ for the buoy, significantly lower than the canonical metabolic quotient of $1.45\pm0.15$. This study shows the added value of co-locating a profiling glider with moored time series buoys, but also demonstrates limitations in achievable precision.



## 1   Introduction

Around a quarter of anthropogenic carbon dioxide ($CO_2$) emitted between 2011 and 2020 was absorbed by the oceans (Friedlingstein et al., 2021). On timescales of less than a day to many months, $CO_2$ in the ocean is influenced by biological (photosynthesis, respiration) and physical processes (air-sea gas exchange, mixing and advection) (Hood and Merlivat, 2001; Takahashi et al., 2002; Copin-Montégut et al., 2004; Hall et al., 2004; Alkire et al., 2014). Understanding the processes affecting export of carbon from the surface to the interior ocean is key for quantifying the effects of a future warmer climate.

Whether a location is predominantly autotrophic (dominated by photosynthesis) or heterotrophic (dominated by respiration) determines the sign of net community production, which is defined as gross primary production (by phytoplankton) minus total respiration (by phytoplankton, zooplankton, and bacteria) (Alkire et al., 2012).

The surface of the Mediterranean Sea ($2.5 \times 10^6$ $km^2$) represents just 0.8 % of oceans globally, but relative to its size, it is regarded as an important sink for anthropogenic carbon dioxide emissions due to higher levels of anthropogenic carbon than in

the Atlantic or Pacific Oceans (Lee et al., 2011; Schneider et al., 2010). This is due to a low Revelle factor related to relatively warm, salty, and high alkalinity waters, encouraging a net flux of carbon dioxide from the atmosphere to the ocean. Carbon dioxide dissolved in water ($CO_2$(aq) and $H_2CO_3$) dissociates to bicarbonate ($HCO_3^-$), and carbonate ($CO_3^{2-}$), releasing $H^+$ ions (Zeebe and Wolf-Gladrow, 2001). $CO_2$(aq), $H_2CO_3$, $HCO_3^-$ and $CO_3^{2-}$ make up total dissolved inorganic carbon (DIC), with $HCO_3^-$ accounting for 90 % of DIC. Carbon dioxide absorbed by the ocean is thought to reach the interior via deep water

formation and biological processes (Álvarez et al., 2014; Arrigo et al., 2008).

The northwestern Mediterranean Sea displays strong seasonal variability. At the surface, temperatures remain at around 13 to 14 °C during winter, increasing to maxima of 26 to 28 °C during summer. Wind-driven vertical mixing occurs during autumn and winter, whilst surface stratification is common during summer as a result of solar heating (Copin-Montégut et al., 2004; D'Ortenzio et al., 2008). Vertical mixing can transport nutrients from greater depths to oligotrophic surface waters (Marty

and Chiavérini, 2002; de Fommervault et al., 2015). A combination of nutrient availability and increased stability driven by surface warming of 0.2 °C can trigger phytoplankton blooms (Yao et al., 2016; Copin-Montégut et al., 2004). The onset of the spring bloom in the northwestern Mediterranean Sea varies from March to April, as observed between 2013 and 2015 at the BOUSSOLE buoy (Bouée pour l'acquisition de Séries Optiques à Long Terme, http://www.obs-vlfr.fr/Boussole/) close to the DyFAMed (Dynamique des Flux Atmospheriques en Mediterraneé) site in the Ligurian-Provençal basin (Merlivat et al.,

2018). Significant increases of particulate (POC) and dissolved organic carbon (DOC) concentrations have been observed during bloom events. POC and DOC export from the ocean surface constitute the 'biological carbon pump' (Carlson et al., 1998; Van Der Loeff et al., 1997; Alkire et al., 2014).

Quantifying these processes in detail requires sufficient data coverage in space and time. Few DIC time series have been maintained continuously, among them the DyFAMed mooring, which is complemented by monthly ship hydrocasts (Copin-

Montégut et al., 2004; Antoine et al., 2008; Taillandier et al., 2012). The DyFAMed site is considered an open ocean location as it is roughly 52 km from the coast in > 2000 m deep water. The mooring is useful for studying processes occurring at specific depth levels at one location (Merlivat et al., 2018; Copin-Montégut et al., 2004; Hood and Merlivat, 2001), but a lack



of vertical and horizontal spatial information is a limiting factor when quantifying mass and energy budgets. Autonomous underwater gliders have been used to survey the northwestern Mediterranean Sea since 2005 (Niewiadomska et al., 2008; Cyr et al., 2017). They are useful platforms for a range of physical and biogeochemical sensors and can operate autonomously for many months in up to 1000 m deep water using battery power (Eriksen et al., 2001; Piterbarg et al., 2014; Queste et al., 2012). The deployment of autonomous platforms, such as underwater gliders, complements fixed-depth time series by enabling observations of biogeochemical and physical horizontal and vertical gradients. This paper aims to estimate net community production ($N$) at the DyFAMed site using in situ continuous measurements from a mooring and an underwater glider deployed in March–April 2016. The additional glider data help overcome limitations of the single-depth mooring data both in terms of vertical data coverage and the contribution of horizontal advection. Furthermore, the glider mission served as a test for a prototype ion-sensitive field-effect transistor (ISFET) pH sensor, which complemented the standard temperature, salinity, and $c(O_2)$ sensors, to provide both $O_2$- and DIC-based net community production.

## 2 Data collection and quality

### 2.1 Ship measurements

Ship CTD and water sample profiles were collected by RV *Téthys II* on 7 March and 16 April 2016 at the DyFAMed site (Fig. 1). Ship hydrocast profiles of temperature and salinity (Sea-Bird Scientific SBE 9 CTD), and $c(O_2)$ (Sea-Bird Scientific SBE 43 sensor), were supplemented by discrete Niskin bottle samples for $c(O_2)$, $c(DIC)$, total alkalinity ($A_T$), and concentrations of total oxidised nitrogen ($NO_x^-$, i.e., the total of $NO_3^-$ and $NO_2^-$), silicate ($Si(OH)_4$), and phosphate ($PO_4^{3-}$), see Appendix A for details. Implausible outliers in the CTD profiles ($< 1\%$ of values) were flagged and excluded from further analysis.

### 2.2 BOUSSOLE and meteorological buoy measurements

At BOUSSOLE, $CO_2$ fugacity $f(CO_2)$ (Wanninkhof and Thoning, 1993) at 10 m depth was measured spectrophotometrically via a CARIOCA sensor using thymol blue pH indicator. Inside an exchanger cell, dissolved $CO_2$ equilibrates with the pH indicator across a silicon membrane. The change in the optical absorbance of the pH indicator is converted to hourly readings of $f(CO_2)$ (Hood and Merlivat, 2001), with an accuracy of 3 µatm (Copin-Montégut et al., 2004). The CARIOCA sensor is replaced roughly every 6 months with a newly calibrated instrument (Merlivat et al., 2018). To remove temperature effects, we show $f_{13}(CO_2)$, normalised to a temperature of $13\,°C$ (Takahashi et al., 1993).

Temperature and salinity were measured hourly using two Sea-Bird Scientific SBE 37-SM MicroCAT sensors at 3 and 10 m depth. The 10 m-MicroCAT failed on 15 March 2016. Up to that point, the mean temperature difference between 10 m and 3 m was $(-0.01 \pm 0.03)\,°C$. The corresponding mean practical salinity $S$(PSS-78) difference was $-0.069 \pm 0.004$. The 10 m-MicroCAT had the same salinity offset to the ship CTD cast on 7 March. The 3 m-MicroCAT agreed to within $0.007\,°C$ with temperature and to within 0.006 with salinity of the ship CTD cast on 7 March. Three days before the ship CTD cast on



**Figure 1.** (a) The northwestern Mediterranean Sea and glider deployment area (small grey box) superimposed on the World Ocean Atlas 2013 surface $c(O_2)$ March climatology (https://www.nodc.noaa.gov/OC5/woa13/woa13data.htm), with accompanying AVISO satellite absolute mean surface currents ($\mathrm{cm\,s^{-1}}$, white arrows) for 6 March–6 April 2016 (http://marine.copernicus.eu/services-portfolio/access-to-products). (b) A close-up of the deployment area at the DyFAMed / BOUSSOLE site. The position of each glider data point (yellow), the DyFAMed mooring (7.90° E, 43.36° N) (white), the BOUSSOLE buoy (7.87° E 43.42° N), and the meteorological buoy 'Côte D'Azur' (7.83° E 43.38° N), are superimposed on surface chlorophyll $a$ concentrations (https://oceancolor.gsfc.nasa.gov/products) on 24 March 2016 (Hu et al., 2012). The bathymetry is flat and greater than 2000 m depth.





16 April, the 3 m-MicroCAT also failed. However, its final salinity reading was within 0.01 of the ship value. We therefore used the 3 m-MicroCAT temperature and salinity readings for calculating buoy-related results.

$c(O_2)$ was measured at BOUSSOLE using Aanderaa oxygen optodes at 3 and 10 m. Only nighttime optode measurements were used because the daytime readings were affected by light-induces spikes (Binetti et al., 2020). Winkler titration samples taken during the ship visits on 7 March and 16 April were used to calibrate optode $c(O_2)$ (Coppola et al., 2018).

The meteorological buoy 'Côte D'Azur' maintained by Météo-France is located close to the BOUSSOLE buoy (Fig. 1). This meteorological buoy measures wind speed at 3.8 m height above sea level (which is extrapolated to 10 m by adding 10 % of the

value), wind direction, air temperature, atmospheric pressure, relative humidity, precipitation and sea surface salinity. Data are archived in the database 'SErvice de DOnnées de l'OMP (SEDOO) Mistrals' (http://mistrals.sedoo.fr/Data-Download). Wind speed and sea level pressure were used in this study.

### 2.3   Glider measurements

An iRobot Seaglider model 1KA (sg537; named 'Fin') with an ogive fairing was deployed close to the DyFAMed mooring

site. A total of 147 dives (294 profiles) were completed by the glider between 7 March and 5 April 2016 covering a diamond-shaped pattern at 7.64–8.00° E, 43.22–43.50° N (Fig. 1). The glider completed 7 circumnavigations of the survey pattern, each in approximately 4 days, with each of the four sides of the pattern taking approximately 1 day to complete. The glider was equipped with a non-pumped SBE model CTD sensor, an Aanderaa model 4330F oxygen optode sensor, a WET Labs Eco Puck sensor measuring optical backscatter at two wavelengths (470 nm and 700 nm), and two paired prototype ISFET pH and

$p(CO_2)$ sensors (Shitashima et al., 2013; Hemming et al., 2017).

Conductivity, temperature, and pressure were used to derive potential temperature ($\theta$) and practical salinity ($S$). Glider $c(O_2)$ was calibrated against ship-based $c(O_2)$ profiles (Fig. 2a, Appendix B1). Outliers ($< 1\%$ of values) outside specified standard deviation ranges (3.5 standard deviations for depths $> 400$ m; 10 standard deviations for depths $< 400$ m) were flagged and discarded from further analysis.

The deployment offered a second opportunity to trial prototype ISFET sensors, previously tested on a glider during the REP14–MED experiment in the Sardinian Sea (Hemming et al., 2017; Onken et al., 2018). ISFET pH was corrected (Fig. 2b) for drift and pressure effects, similar to the steps undertaken by Hemming et al. (2017). The drift- and pressure-corrected ISFET pH was calibrated by linear regression against ship pH on 7 March and 16 April 2016 (B2, C).

### 3   Development of a spring bloom

The spring bloom at the DyFAMed site is characterised by a decrease of surface $f_{13}(CO_2)$ due to photosynthesis. Its onset varies from year to yer (e.g., April 2013, March 2014), and usually occurs after a period of deep mixing (Merlivat et al., 2018). The glider deployment in March–April 2016 provided horizontal and vertical context to the fixed-depth near-surface mooring sensors when the spring bloom was expected to occur.



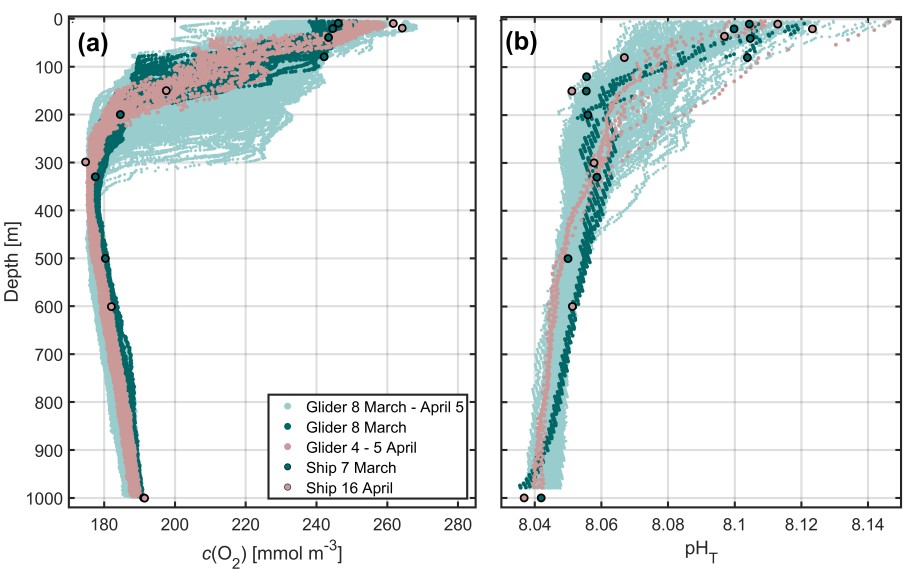

**Figure 2.** Calibrated glider (a) $c(O_2)$ and (b) ISFET $pH_T$ over the whole deployment (pale green), 7–8 March (dark green), 4–5 April (pink), and the ship Winkler samples collected on 7 March (dark green circles), and on 16 April (pink circles).

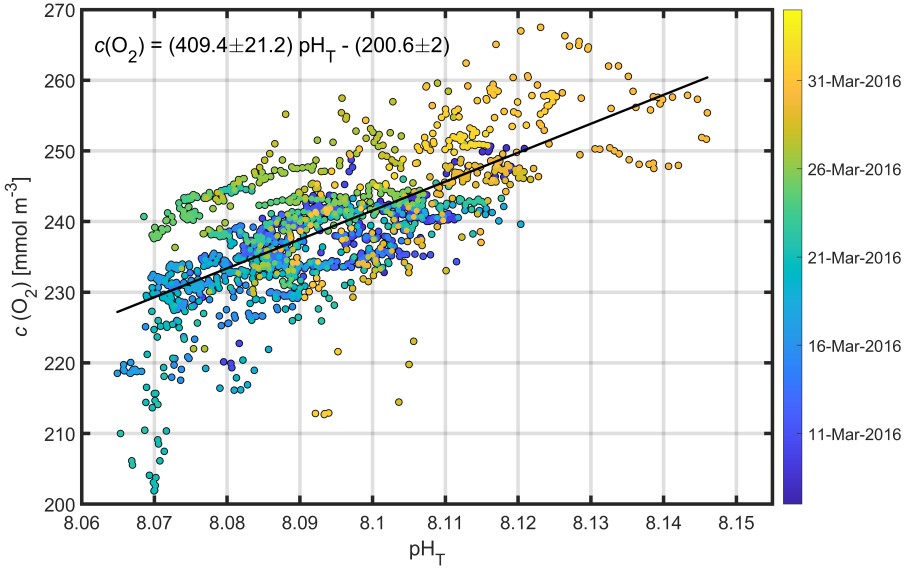

**Figure 3.** Glider $pH_T$ against glider $c(O_2)$, and coloured by time for all points within the mean euphotic layer (top 46 m). The linear fit and corresponding equation between $pH_T$-8 and $c(O_2)$ is also shown.



Sea surface temperature (SST) remained relatively stable below $13.5\,°C$ between 7 March and 19 March whilst surface
$f_{13}(CO_2)$ increased by 40 µatm (Fig. 4a), and surface salinity increased by 0.13 (Fig. 4b). The higher $f_{13}(CO_2)$, higher
salinity waters were likely the results of wind-induced deep mixing and increased convection (Fig. 4a). These waters originate
from 50–150 m depth where high salinity (Fig. 5b), high $f_{13}(CO_2)$, low $c(O_2)$ (Fig. 5c) Levantine Intermediate Water (LIW)
exists (Knoll et al., 2017). SST increased by up to $0.6\,°C$ between 19 and 20 March, and continued to increase intermittently
to a maximum of $14.3\,°C$ on 5 April (total increase of $1\,°C$ over the deployment period).

Whilst SST increased between 19 March and 1 April, $f_{13}(CO_2)$ decreased by 85 µatm. This was the start of the spring bloom,
associated with increased surface $c(O_2)$, $pH_T$, and optical backscatter (Fig. 4b, Fig. 5c,d,e). $O_2$ is produced by photosynthesis.
The $pH_T$ increase reflects biological $CO_2$ uptake altering the carbonate equilibria (Cornwall et al., 2013; Copin-Montégut and
Bégovic, 2002), and increases in temperature. Optical backscatter relates to predominantly particulate organic matter (Stramski
et al., 2004), with minor contributions from mineral particles and gas bubbles at this site. Therefore, all of these observations
suggested an increase in biological production.

A clear relationship between potential temperature $(θ)$, $c(O_2)$, and $pH_T$ existed within the euphotic layer (mean depth:
$z_{eu} = 46±7\,m$) (Figs. 3, 5a,c,e), with higher potential temperature associated with higher $pH_T$ and $c(O_2)$. The stronger surface
stratification resulting from calmer meteorological conditions, (e.g. reduced wind speed after 19 March, Fig. 4a), enhancing
light supply and stability, causing productivity to increase (Sverdrup, 1953; Pingree et al., 1977). As a result of increased
photosynthesis, surface waters became $O_2$-supersaturated by 27 March (Fig. 4b, Fig. 5c).

Nutrient concentrations obtained by ship hydrocasts for depths of 10–90 m on 7 March were generally highest at 90 m
where there is increased remineralisation, and lowest at the surface where there is increased usage by phytoplankton (Fig. 6a).
Clear differences can be seen in nutrient samples obtained before and after the spring bloom (Fig. 6b). $NO_x^-$ concentrations
decreased by $7\,mmol\,m^{-3}$, $Si(OH)_4$ by $2.5\,mmol\,m^{-3}$ and $PO_4^{3-}$ by $0.22\,mmol\,m^{-3}$ at 30 m depth, associated with bio-
logical production during the spring bloom period. A deep chlorophyll $a$ maximum is often found around this depth (Estrada,
1996) later in the season but was not apparent in our data. At 70 m and 90 m depth, nutrient concentrations did not vary much
between March and April, while there was a significant decrease in $c(O_2)$ and a small change in potential temperature (Fig. 6c).
This corresponded to increased stratification with a stronger potential temperature gradient within the top 70 m.

## 4  Estimating net community production from $O_2$ and DIC mass budgets

Net community production $N$ is estimated based on mass budgets using the method described by (Alkire et al., 2014):

$$\int_0^{z_{lim}} \left( \frac{\partial c}{\partial t} + u\frac{\partial c}{\partial x} + v\frac{\partial c}{\partial y} \right) \mathrm{d}z = -F_{ase} + N(O_2) + F_e + F_v \tag{1}$$

This equation applies to $N(O_2)$. For DIC, the $N(O_2)$ term is replaced with $-N(DIC)$ because during photosynthesis, DIC
is consumed and $O_2$ is produced, and vice versa for respiration.





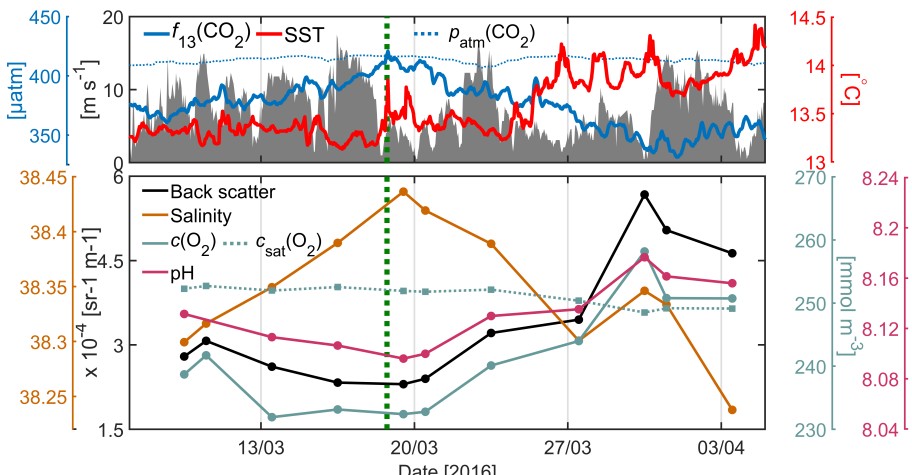

**Figure 4.** (a) Wind speed at 10 m above sea level at the meteorological buoy (dark grey shading), sea surface temperature (red), and $CO_2$ fugacity normalised to $13\,^{\circ}C$ (blue) at 10 m depth at the BOUSSOLE buoy. The partial pressure of atmospheric $CO_2$ in March 2016 (dotted blue) at Lampedusa, Italy, is shown. The onset of the spring bloom (dashed green), determined using buoy $f(CO_2)$ measurements, is plotted for comparison.(b) Glider measurements of salinity (orange), optical backscatter at 700 nm (black), dissolved oxygen concentrations ($c(O_2)$) (turquoise), and calibrated $pH_T$ (pink) binned at 10 m. The oxygen concentration at saturation (dashed turquoise; parameterisation of García and Gordon (1992) based on Benson and Krause (1984)) is plotted for comparison.

The following diagnostic equations are therefore used to calculate $N$ based on $O_2$ and DIC mass budgets:

$$N(O_2) = \frac{\Delta I(O_2)}{\Delta t} + F_{\mathrm{ase}}(O_2) + F_{\mathrm{adv}}(O_2) - F_{\mathrm{e}}(O_2) - F_{\mathrm{v}}(O_2) \tag{2}$$

$$N(\mathrm{DIC}) = -\frac{\Delta I(\mathrm{DIC})}{\Delta t} - F_{\mathrm{ase}}(\mathrm{DIC}) - F_{\mathrm{adv}}(\mathrm{DIC}) + F_{\mathrm{e}}(\mathrm{DIC}) + F_{\mathrm{v}}(\mathrm{DIC}) \tag{3}$$

$\frac{\Delta I}{\Delta t}$ is the integrated column inventory change over time, $F_{\mathrm{adv}}$ is horizontal advection, $F_{\mathrm{ase}}$ is the air-sea flux (positive for the direction ocean to atmosphere), $F_{\mathrm{e}}$ is the entrainment flux due to mixed layer deepening, and $F_{\mathrm{v}}$ is the diapycnal eddy diffusion flux. The time period chosen to derive the fluxes required to calculate $N$ spanned 25 days between 10 March and 3 April, and included 285 individual profiles (glider dives 10 to 147) spanning the entire survey domain. The first nine dives were omitted because they were used for glider flight trimming. Mean values over the 25 day-period are shown in Table 2. Table 1 gives the uncertainties considerd for each budget term.

## 4.1 Inventory change $\frac{\Delta I}{\Delta t}$

Daily mean column inventories were integrated between surface and $z_{\mathrm{lim}}$ for all glider profiles within a 4-day moving window centred on each date. As integration depth $z_{\mathrm{lim}}$, we used the mean euphotic depth of 46 m, derived from satellite measurements



**Figure 5.** Glider (a) potential temperature ($\theta$), (b) salinity, (c) dissolved oxygen concentration ($c(O_2)$), (d) optical backscatter, (e) pH$_T$, and (f) dissolved inorganic carbon concentration ($c(DIC)$). The mean euphotic depth used for calculating $N$ (green dashed line, $z_{lim} = 46\,m$) and mixed layer depth (white line; derived using the potential density criteria of the hybrid scheme of Holte and Talley (2009)) are superimposed. The vertical dashed black lines represent the glider reaching the most northerly point of its transects.





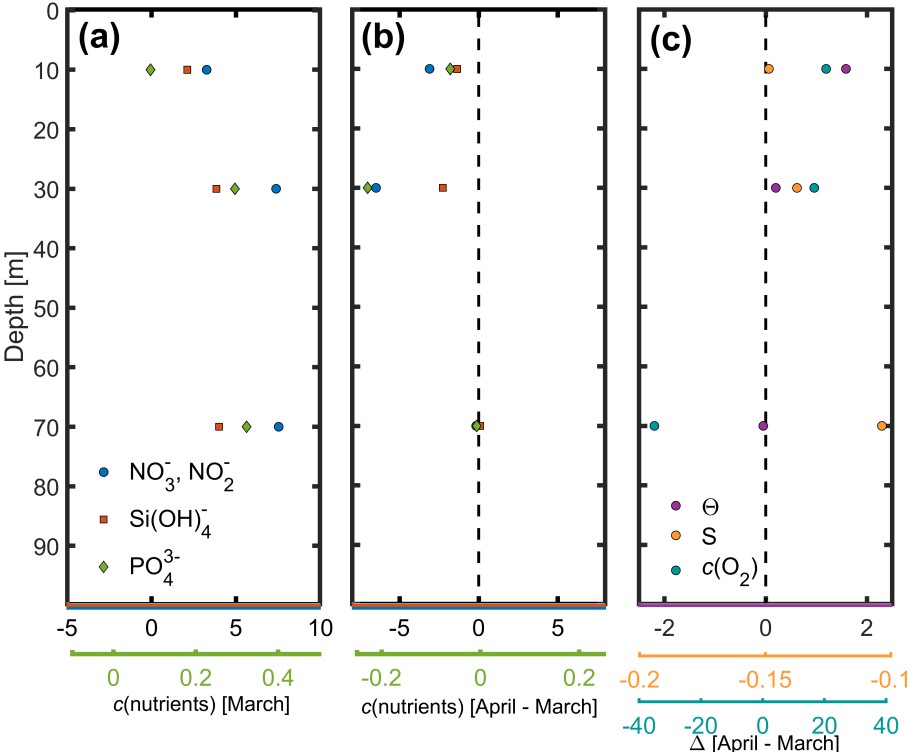

**Figure 6.** (a) Nutrient concentrations (mmol m$^{-3}$) in March 2016, (b) change in nutrient concentrations between March and April 2016, and (c) changes in potential temperature ($\theta$), practical salinity ($S$), and $c(O_2)$ (mmol m$^{-3}$) between March and April 2016.

using the method described by Lee et al. (2007). Inventory changes ($\frac{\Delta I}{\Delta t}$) were calculated from the day-to-day inventory differences:

$$\frac{\Delta I}{\Delta t} = \frac{\int_0^{z_{\text{lim}}} c_{t+1\,\text{d}}(z)\text{d}z - \int_0^{z_{\text{lim}}} c_t(z)\text{d}z}{1\,\text{d}} \qquad (4)$$

### 4.2 Advection $F_{\text{adv}}$

The glider measured within an Eulerian framework, sampling geographic locations over time. In contrast to a Lagrangian framework, estimates of $F_{\text{adv}}$ are needed to close the budget. Advection was calculated following Alkire et al. (2014) using zonal and meridional mean horizontal concentration gradients and current velocities within $z_{\text{lim}}$ using a moving time window of 8 days. This longer time window of 8 days was chosen to reduce the error in the mean gradients (Fig. 7). Mean horizontal





**Table 1.** The uncertainties considered for each flux term when estimating total propagated error. The mean errors for $N$ using glider ($N_\mathrm{g}$) and buoy ($N_\mathrm{b}$) measurements are listed for both $c(\mathrm{O_2})$ and $c(\mathrm{DIC})$ in $\mathrm{mmol\,m^{-2}\,d^{-1}}$.

| Term | Uncertainties considered | $N_\mathrm{g}(\mathrm{O_2})$ | $N_\mathrm{b}(\mathrm{O_2})$ | $N_\mathrm{g}(\mathrm{DIC})$ | $N_\mathrm{b}(\mathrm{DIC})$ |
|---|---|---|---|---|---|
| $\frac{\Delta I}{\Delta t}$ | Standard error of profiles (glider) or concentrations over time (buoy); root mean square error of the calibration fits (glider) | 14 | 9 | 38 | 8 |
| $F_\mathrm{adv}$ | 10 % absolute velocity; concentration and velocity plane-fit errors | 29 | 29 | 85 | 85 |
| $F_\mathrm{ase}$ | Root mean square error of the calibration fits ($\mathrm{O_2}$); 20 % uncertainty of the gas transfer velocity (Wanninkhof, 2014); standard deviations of temperature and salinity when calculating $c_\mathrm{sat}$ | 15 | 14 | 2 | 2 |
| $F_\mathrm{e}$ | Standard error of inventory | 0.3 | 0.3 | 0.6 | 0.6 |
| $F_\mathrm{v}$ | Standard error of $F_\mathrm{v}$ over 4-day moving window | 0.2 | 0.2 | 0.1 | 0.1 |

concentration gradients $A_c(z)$ and $B_c(z)$ were calculated for each 5 m depth bin using robust plane-fits of all concentration

measurements in the 8-day time window:

$$c_\mathrm{fit}(z) = A_c(z)x + B_c(z)y + d_c(z) \qquad (5)$$

where $d_c(z)$ is a constant (not used), and $x$ and $y$ are Cartesian coordinates calculated using zonal and meridional distances, respectively. Bisquare weights were applied when determining the fit coefficients (Gross, 1977) to limit the effect of outliers.

The zonal (east-west) concentration gradients were small for both $\mathrm{O_2}$ and DIC (Fig. 8). Opposing meridional (north-south)

gradients for $\mathrm{O_2}$ and DIC indicate biological patchiness. However, interestingly the positive gradient for $\mathrm{O_2}$ (corresponding to higher concentrations in the north) is opposite to the pattern seen in the satellite-derived ocean colour (1b), which shows lower chlorophyll $a$ concentrations in the north (implying that the bloom propagates from south to north). This highlights that primary production and net community production are not always directly correlated.

To estimate the absolute geostrophic currents within the survey domain the dynamic height anomaly ($\Psi$) was calculated

relative to the surface using glider salinity, temperature and pressure (Roquet et al., 2015), and planes were fitted for a moving time window of 8-days:

$$\Psi_\mathrm{fit}(z) = A_\Psi(z)x + B_\Psi(z)y + d_\Psi(z) \qquad (6)$$

Using these plane-fits, the meridional and zonal components of geostrophic shear were derived and referenced to the 8-day running means of the meridional and zonal components of OSCAR velocities (Bonjean and Lagerloef, 2002). Referencing

the geostrophic shear with glider dive-averaged currents (DACs) was initially explored, but when compared with satellite data products (Fig. 8c,d) it was clear that the meridional velocities ($v$) were unrealistic (cf. section 7).





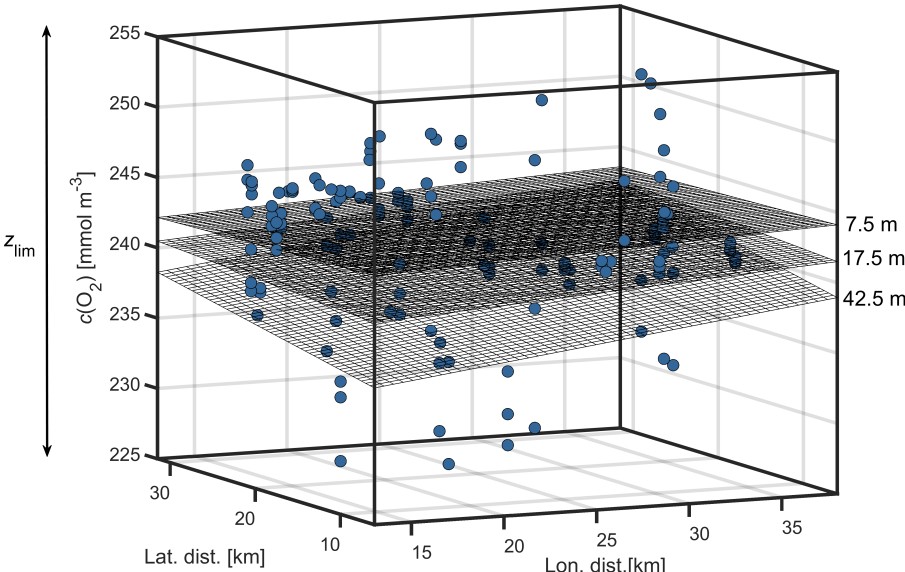

**Figure 7.** Plane-fits of dissolved oxygen concentrations ($c(O_2)$) at 7.5 m), 17.5 m and 42.5 m depth within $z_{lim}$ for 25 March 2016.

The advection flux ($F_{adv}$) was calculated from the horizontal concentration gradients and the absolute geostrophic velocities:

$$F_{adv}(z) = A_c(z)u(z) + B_c(z)v(z) \tag{7}$$

The mean gradients using measurements binned every 5 m between the surface and $z_{lim}$ were averaged at each time step and used for estimating advection (Fig. 8a,b).

### 4.3 Air-sea gas exchange $F_{ase}$

Air-sea gas exchange (negative for fluxes from atmosphere to ocean) was estimated for $O_2$ and $CO_2$/DIC using:

$$F_{ase}(O_2) = k(O_2) \left[ c_{surf}(O_2) - (1 + \Delta_{bubble}) c_{sat}(O_2) \right] \frac{\min(z_{lim}, z_{mix})}{z_{mix}} \tag{8}$$

with the equilibrium bubble correction $\Delta_{bubble} = 0.01[w_{10}/(9\,\mathrm{m\,s^{-2}})]^2$ (Woolf and Thorpe, 1991), and

$$F_{ase}(CO_2) = k(CO_2) \left[ c_{surf}(CO_2) - c_{sat}(CO_2) \right] \frac{\min(z_{lim}, z_{mix})}{z_{mix}} \tag{9}$$

The gas transfer velocity $k$ is calculated according to Wanninkhof (2014). $c_{surf}$ is the mean concentration in the top 10 m over a moving 4-day time window. The mixed layer depth $z_{mix}$ was estimated using the potential density criteria of the algorithm, of Holte and Talley (2009), with a threshold of 0.03 $\mathrm{kg\,m^{-3}}$ and gradient of 0.0005 $\mathrm{kg\,m^{-3}\,dbar^{-1}}$. These criteria are sensitive




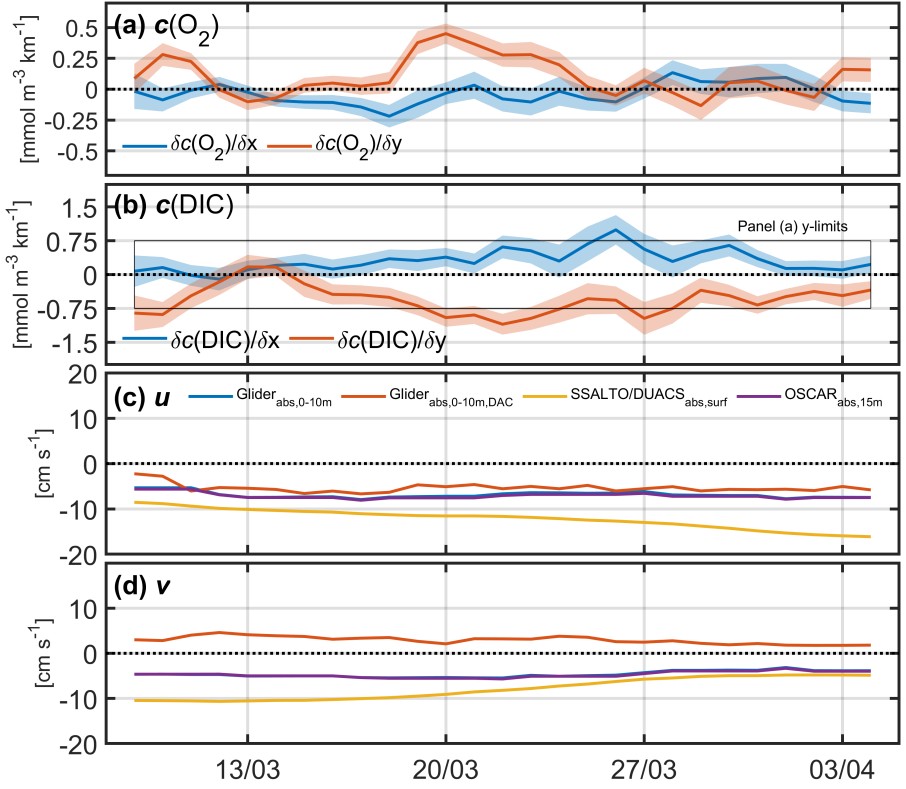

**Figure 8.** Zonal and meridional mean horizontal concentration gradients for (a) $c(O_2)$ and (b) $c(DIC)$ estimated using robust plane-fits. The black box in panel (b) highlights the y-limits for panel (a) for reference. (c) Absolute mean $u$ velocity, and (d) $v$ velocity, using glider measurements in the top 10 m referenced to OSCAR velocities (blue), and referenced to dive-averaged currents (DACs, red), absolute surface mean currents from satellite products SSALTO/DUACS (yellow) and OSCAR (purple).

enough to detect the actual mixing layer-depth relevant for gas exchange fluxes (Brainerd and Gregg, 1995). $c_{sat}(O_2)$ is

the $O_2$ air-saturation concentration of García and Gordon (1992), corrected to actual sea-level pressure, $p_{baro}$. $c_{sat}(CO_2) = \chi(CO_2)p_{baro}F(CO_2)$ is the $CO_2$ air-saturation concentration, calculated using the dry mole fraction of atmospheric $CO_2$, $\chi(CO_2)$, at the nearest NOAA Carbon Cycle network station Lampedusa, Italy, in March 2016 (ftp://aftp.cmdl.noaa.gov/data/trace_gases/co2/flask/surface, site code: LMP, Dlugokencky et al. (2021)). $F(CO_2)$ is the $CO_2$ solubility function in mol dm$^{-3}$ atm$^{-1}$ (Weiss and Price, 1980). The last term of the equations is a scaling factor that apportions the air-sea exchange

flux to the depth layer of interest ($z_{lim} = 46$ m) when $z_{mix}$ is deeper than $z_{lim}$.



### 4.4 Entrainment $F_{\mathrm{e}}$

When the mixed layer deepens to depths greater than $z_{\mathrm{lim}}$, water from below with different concentrations mix with water within the layer. As a result, there is either a reduction or an increase in mean concentration within $z_{\mathrm{lim}}$ depending on the vertical concentration distribution. To account for this, $F_{\mathrm{e}}$ was estimated following Binetti et al. (2020):

$$F_{\mathrm{e}} = \frac{I_1(z_{\mathrm{mix}}(t_2))\frac{z_{\mathrm{lim}}}{z_{\mathrm{mix}}(t_2)} - I_1(z_{\mathrm{lim}})}{t_2 - t_1} \tag{10}$$

where $I_1(z_{\mathrm{mix}}(t_2))$ is the inventory at time $t_1$ within the mixed layer depth $z_{\mathrm{mix}}$ at time $t_2$; $I_1(z_{\mathrm{lim}})$ is the inventory at $t_1$ within $z_{\mathrm{lim}}$; $t_2 - t_1$ represents the time difference. Mean values of $z_{\mathrm{mix}}$ and $c$ averaged over a 4 day moving time window were used.

The mean $F_{\mathrm{e}}(O_2)$ was $(-65 \pm 1)\,\mathrm{mmol\,m^{-2}\,d^{-1}}$ and the mean $F_{\mathrm{e}}(\mathrm{DIC})$ was $(-74 \pm 1)\,\mathrm{mmol\,m^{-2}\,d^{-1}}$ for the 7 days in the 25 day-period when entrainment (mixed-layer deepening) occurred and $z_{\mathrm{mix}} > z_{\mathrm{lim}}$.

### 4.5 Vertical diapycnal eddy diffusion $F_{\mathrm{v}}$

Vertical diapycnal eddy diffusion across the base of $z_{\mathrm{lim}}$ was estimated using the measured concentration–density gradients (Copin-Montégut, 2000) and an estimate of the turbulence energy dissipation rate $\epsilon$:

$$F_{\mathrm{v}} = \frac{0.25\epsilon\rho}{g}\frac{\partial c}{\partial \rho} \tag{11}$$

where the concentration gradient was evaluated using centred finite differences at $z_{\mathrm{lim}} \pm 5\,\mathrm{m}$. $\epsilon = 1.5 \times 10^{-9}\,\mathrm{m^2\,s^{-3}}$ was taken from measurements in the Mediterranean bottom pycnoline (Cuypers et al., 2012). Copin-Montégut (2000) guessed a 33 times higher value of $5 \times 10^{-8}\,\mathrm{m^2\,s^{-3}}$ in her study of net community production at DYFAMED, but this value was probably an overestimate because even in the presence of eddies, Cuypers et al. (2012) estimates $\epsilon$ to be only $8.5 \times 10^{-9}\,\mathrm{m^2\,s^{-3}}$. Concentration gradients were calculated for each profile within a 4-day moving window and then averaged to get one gradient per time step.

The mean $F_{\mathrm{v}}(O_2)$ was $(-0.8 \pm 0.2)\,\mathrm{mmol\,m^{-2}\,d^{-1}}$ and the mean $F_{\mathrm{v}}(\mathrm{DIC})$ was $(-0.1 \pm 0.1)\,\mathrm{mmol\,m^{-2}\,d^{-1}}$ over the 25 day-period. With the higher $\epsilon$ value of $8.5 \times 10^{-9}\,\mathrm{m^2\,s^{-3}}$ found in eddies, these fluxes would be 5.7 times higher, but still negligible with respect to the other four terms in the mass budget.

### 4.6 Buoy measurements

We calculate $N$ from $c(O_2)$ and $c(\mathrm{DIC})$ at the BOUSSOLE buoy ($N_{\mathrm{b}}$) for comparison with $N_{\mathrm{g}}$ estimates. $N_{\mathrm{b}}$ was estimated similarly to $N_{\mathrm{g}}$, in that we follow the budget approach defined in Sect. 4, and equations defined thereafter. Inventory changes were calculated from surface observations multiplied by $z_{\mathrm{lim}}$ because depth-profiles are not measured at the buoy. When $z_{\mathrm{mix}} < z_{\mathrm{lim}}$, this most likely results in an overestimate of the actual inventory change because $N$ decreases with depth. Also, $F_{\mathrm{ase}}$ was scaled in the same way as the glider-based estimates when $z_{\mathrm{mix}} > z_{\mathrm{lim}}$. Finally, $F_{\mathrm{adv}}$, $F_{\mathrm{e}}$, and $F_{\mathrm{v}}$ could not be derived from the single-depth measurements at BOUSSOLE. Instead, the glider-derived fluxes were also used to compute $N_{\mathrm{b}}$.




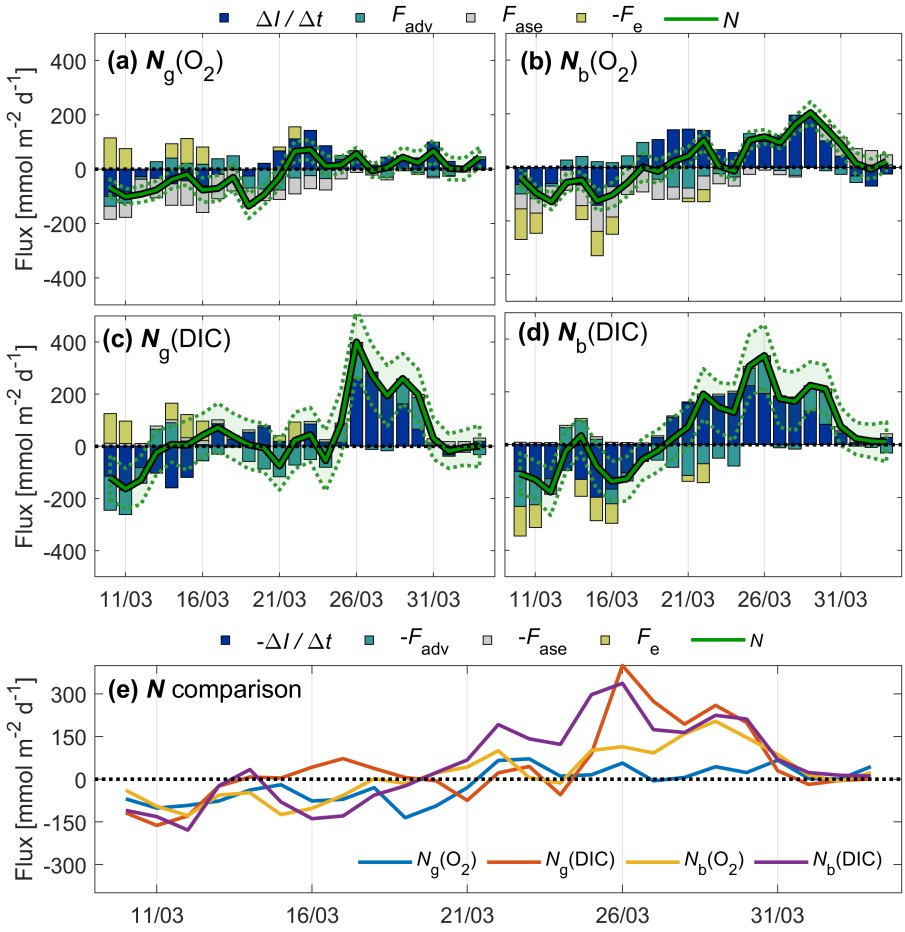

**Figure 9.** Inventory change ($\Delta I/\Delta t$), advection ($F_{\mathrm{adv}}$), air-sea exchange ($F_{\mathrm{ase}}$), entrainment ($F_{\mathrm{e}}$) fluxes and net community production ($N$) of $O_2$ (a,b; top legend) and DIC (c,d; bottom legend) related to glider (a,c) and buoy (b,d) measurements. Uncertainty intervals are shown as dotted lines. All $N$ estimates are compared in panel (e).

## 5 Net community production $N$

Prior to 21 March, $N_{\mathrm{g}}$ and $N_{\mathrm{b}}$ were negative or close to zero most of the time. This could be a sign of local net heterotrophic conditions, but could also be due to underestimated entrainment ($F_{\mathrm{e}}$) or vertical eddy diffusion fluxes ($F_{\mathrm{v}}$). This observation cannot be explained by physical undersaturation driven by recent cooling because temperatures decreased by only $0.2\,°\mathrm{C}$ between 1 and 21 March, explaining at most $0.4\,\%$ undersaturation. After 21 March, $N$ exceeds zero until the end of the deployment period. The approximate start of the spring bloom was 19 March as evidenced by the buoy $c(\mathrm{DIC})$ (Fig. 4), which is also the time when $N_{\mathrm{b}}(\mathrm{DIC})$ becomes positive, lasting until the end of the deployment period.



**Table 2.** Comparison of $\Delta I/\Delta t$, $F_{\mathrm{adv}}$, $F_{\mathrm{ase}}$, $F_{\mathrm{v}}$ and $N$ estimates using $c(O_2)$ and $c(DIC)$, and the corresponding flux terms used in this study with others in the literature. The $N$ estimates of Copin-Montégut (2000) are based on $\Delta I/\Delta t$, $F_{\mathrm{ase}}$, and $F_{\mathrm{v}}$ (calculated with a turbulence energy dissipation rate of $\epsilon = 5.0 \times 10^{-8}\,\mathrm{m^2\,s^{-3}}$).

| Time period | Variable | $\Delta I/\Delta t$ | $F_{\mathrm{adv}}$ | $F_{\mathrm{ase}}$ | $F_{\mathrm{v}}$ | $N$ |
|---|---|---|---|---|---|---|
| | | | | $\mathrm{mmol\,m^{-2}\,d^{-1}}$ | | |
| 10 March – 3 April 2016 | $O_2$ glider | $15 \pm 14$ | $-9 \pm 29$ | $-42 \pm 15$ | $-1 \pm 0$ | $-17 \pm 36$ |
| | $O_2$ buoy | $37 \pm 9$ | $-9 \pm 29$ | $-30 \pm 14$ | $-1 \pm 0$ | $17 \pm 37$ |
| | DIC glider | $-24 \pm 38$ | $12 \pm 85$ | $-10 \pm 2$ | $1 \pm 0$ | $44 \pm 94$ |
| | DIC buoy | $-32 \pm 8$ | $12 \pm 85$ | $-9 \pm 2$ | $1 \pm 0$ | $49 \pm 86$ |
| 19 March – 3 April 2016 | $O_2$ glider | $49 \pm 15$ | $-21 \pm 31$ | $-24 \pm 9$ | $-1 \pm 0$ | $9 \pm 36$ |
| | $O_2$ buoy | $85 \pm 11$ | $-21 \pm 31$ | $-2 \pm 7$ | $-1 \pm 0$ | $67 \pm 35$ |
| | DIC glider | $-77 \pm 41$ | $7 \pm 89$ | $-9 \pm 2$ | $1 \pm 0$ | $85 \pm 98$ |
| | DIC buoy | $-120 \pm 7$ | $7 \pm 89$ | $-9 \pm 2$ | $1 \pm 0$ | $128 \pm 90$ |
| Coppola et al. (2018) | | | | | | |
| 1994 – 2014 average | $O_2$ ship | — | — | $7$ | — | $25$ |
| 1994 – 2014 monthly averages | $O_2$ ship | — | — | –41 to 40 | — | — |
| Copin-Montégut (2000) | | | | | | |
| 4–8 May 1995 (0–30 m) | $O_2$ ship | $-3 \pm 14$ | — | $20 \pm 6$ | $-26 \pm 10$ | $43 \pm 19$ |
| 4–8 May 1995 (0–38 m) | DIC ship | $-28 \pm 13$ | — | $-1 \pm 0$ | $39 \pm 4$ | $68 \pm 13$ |
| 19–23 May 1995 (0–46 m) | $O_2$ ship | $56 \pm 18$ | — | $14 \pm 4$ | $-45 \pm 10$ | $115 \pm 23$ |
| 19–23 May 1995 (0–46 m) | DIC ship | $-36 \pm 18$ | — | $-2 \pm 1$ | $31 \pm 4$ | $69 \pm 19$ |

$N_{\mathrm{g}}(O_2)$ peaks on 22–23 and 31 March at around $70\,\mathrm{mmol\,m^{-2}\,d^{-1}}$, matching periods of high $c(O_2)$ (Fig. 5c). In contrast, $N_{\mathrm{b}}(O_2)$ is highest between 25 and 31 March at $\geq 80\,\mathrm{mmol\,m^{-2}\,d^{-1}}$. The total range of $N(DIC)$ is larger than for $N(O_2)$. $N_{\mathrm{g}}(DIC)$ and $N_{\mathrm{b}}(DIC)$ reach similar peak values on 26 and 29–30 March.

The contributions of $F_{\mathrm{adv}}$ and $F_{\mathrm{e}}$ are significant at times. $F_{\mathrm{e}}(O_2) < -80\,\mathrm{mmol\,m^{-2}\,d^{-1}}$ on days when wind is high and the surface mixed layer is deepening (Fig. 9), and there are periods when absolute $F_{\mathrm{adv}}(O_2) > 50\,\mathrm{mmol\,m^{-2}\,d^{-1}}$. Absolute $F_{\mathrm{adv}}(DIC)$ is on average $34\,\mathrm{mmol\,m^{-2}\,d^{-1}}$ higher than $F_{\mathrm{adv}}(O_2)$, and on some days absolute $F_{\mathrm{adv}}(DIC) > 100$ $\mathrm{mmol\,m^{-2}\,d^{-1}}$. $F_{\mathrm{e}}(DIC)$ is generally of absolute $F_{\mathrm{e}}(O_2)$. Surface waters are undersaturated with $O_2$ prior to 29 March (negative $F_{\mathrm{ase}}$), and then oversaturated until the end of the deployment period (positive $F_{\mathrm{ase}}$; Fig. 4.) Surface waters are
undersaturated in $CO_2$ throughout the deployment period (negative $F_{\mathrm{ase}}$).

Between 10 March and 3 April we estimate an average $N$ of between $(-17 \pm 35)$ and $(49 \pm 86)\,\mathrm{mmol\,m^{-2}\,d^{-1}}$, depending on the variable used (Table 2). Between 19 March and 3 April, $N$ was positive most of the time, and on average ranged between $(9 \pm 36)$ and $(128 \pm 90)\,\mathrm{mmol\,m^{-2}\,d^{-1}}$.



## 6 Stoichiometric relationship

Dividing $N(O_2)$ by $N(DIC)$ provides the photosynthetic quotient $Q_P$, previously estimated as $1.45 \pm 0.15$ (Laws, 1991; Anderson and Sarmiento, 1995; Anderson, 1995). The average $Q_P$ in this study using $N_g(O_2)$ and $N_g(DIC)$ between 10 March and 3 April was $0.14 \pm 0.81$, excluding periods when $N_g(DIC) < 30 \, \mathrm{mmol \, m^{-2} \, d^{-1}}$ (i.e., close to zero within its uncertainty). When using $N_b(O_2)$ and $N_b(DIC)$ over the same time period , $Q_P$ was on average $0.49 \pm 0.60$. Using linear regression of $N(O_2)$ against $N(DIC)$ provides an alternative way to derive $Q_P$. The resulting slopes are $0.25 \pm 0.06$ for the glider and $0.54 \pm 0.06$

for the buoy $N$ estimates. These $Q_P$ values therefore do not match the canonical values of $1.45 \pm 0.15$, but similarly low values of $0.63 \pm 0.30$ were observed by Copin-Montégut (2000) in early May 1995 at the same site (2)

   The change of nutrient concentrations provides an alternative way of estimating net community production. Assuming the observed $NO_x^-$ drawdown of 7 $\mathrm{mmol \, m^{-3}}$ and the $PO_4^{3-}$ drawdown of 0.22 $\mathrm{mmol \, m^{-3}}$ occurred over a period of 15 to 25 days (with 15 days corresponding to the period 19 March to 3 April; the exact period is not known because we only have

observations on 7 March and 16 April) and extends over the same depth horizon (46 m) as used for the $O_2$- and DIC-based $N$ calculations, and neglecting any additional contribution to NCP fuelled by diapycnal mixing, this corresponds to rates of 13 to 21 $\mathrm{mmol \, mmol \, m^{-2} \, d^{-1}}$ for nitrogen and 0.4 to 0.7 $\mathrm{mmol \, mmol \, m^{-2} \, d^{-1}}$. Assuming a Redfield C:N:P stoichiometric ratio of 106:16:1 (Redfield et al., 1963), this corresponds to carbon fluxes of 85 to 142 $\mathrm{mmol \, m^{-2} \, d^{-1}}$ (based on nitrogen) and 43 to 72 $\mathrm{mmol \, m^{-2} \, d^{-1}}$. The values are in reasonable agreement with the $N(DIC)$ values derived for the bloom period between 19

March and 3 April, indicating nutrient consumption in line with the assumed stoichiometric ratio. However, the N-based value is far higher, indicating that there is less carbon fixed than one would expect assuming Redfield stoichiometry. The discrepancy between expected and observed stoichiometry is even larger for the $O_2$-based $N$ values.

   As mentioned, Copin-Montégut (2000) also found non-Redfield ratios for $O_2$ and DIC-derived $N$ at the same DYFAMED site. We do not have a physiological explanation for such large deviations from the canonical stoichiometric values. It is

unlikely to be due to a calibration error in the $O_2$ concentration because it would have to be of the order of 20 $\mathrm{mmol \, m^{-3}}$ and our observations at depths > 400 m agree to within 3 $\mathrm{mmol \, m^{-3}}$ with the record of historic measurements.

## 7 Discussion

Quantifying net community production helps us to understand the role of biological production and consumption on carbon export from the surface to deep waters. If the rate of carbon export is known, the rate of atmospheric $CO_2$ drawdown into

the ocean can be better quantified, and the accuracy of future climate projections might improve. Using underwater gliders as tools to observe the water column on timescales of less than a month over a wider area allow us to estimate physical processes (e.g. mixing, advection) that affect biogeochemical tracer concentrations and estimates of net community production derived therefrom.

   During the spring bloom we estimate $N$ between $(-22 \pm 31)$ and $(119 \pm 88) \, \mathrm{mmol \, m^{-2} \, d^{-1}}$ on average, with maxima

> 300 $\mathrm{mmol \, m^{-2} \, d^{-1}}$ at times. Few studies have estimated $N$ at the DyFAMed site, and there are currently no $N$ estimates that incorporate $F_{adv}$ using concentrations over a larger area surrounding DyFAMed.



Another study estimated an annual mean $N(O_2)$ of $9.2 \, \mathrm{mol\,m^{-2}\,a^{-1}}$ (equivalent to $25 \, \mathrm{mmol\,m^{-2}\,d^{-1}}$), and monthly mean $O_2$ production of between $11 \, \mathrm{mmol\,m^{-2}\,d^{-1}}$ and $19 \, \mathrm{mmol\,m^{-2}\,d^{-1}}$ in March–April, using excess $O_2$ above $100\,\%$ saturation over a period of 20 years (Coppola et al., 2018). Further, Copin-Montégut (2000) estimated $N(O_2)$ as $(43\pm19) \, \mathrm{mmol\,m^{-2}\,d^{-1}}$

in the top 30 m during 4–8 May 1995, and as $(115\pm23) \, \mathrm{mmol\,m^{-2}\,d^{-1}}$ in the top 46 m during 19–23 May 1995. A direct comparison is not possible between this study, Coppola et al. (2018), and Copin-Montégut (2000) because each study is focused on different timescales (from years to days) or different seasons. Keeping this in mind, our range of $N$ estimates are similar to those estimated by Coppola et al. (2018) and Copin-Montégut (2000).

Uncertainties for $N(\mathrm{DIC})$ are generally higher than those for $N(O_2)$, and the highest uncertainties are associated with

$F_{\mathrm{adv}}$. $F_{\mathrm{adv}}$ uncertainties are high because the uncertainties associated with the $c$ plane-fits are high due to $c(\mathrm{DIC})$ variability within the time-centred window. For example, $c(\mathrm{DIC})$ within $z_{\mathrm{lim}}$ varied over $33 \, \mathrm{mmol\,m^{-3}}$ for the time-centred window on 16 March leading to low standard errors (Fig. 8b), while $c(\mathrm{DIC})$ varied over $40 \, \mathrm{mmol\,m^{-3}}$ for the time-centred window on 26 March leading to high standard errors. For this latter window centred on 26 March, the decrease in $c(\mathrm{DIC})$ over time between 25 and 30 March caused this high uncertainty. We use an 8-day time-centred moving window to reduce the potential

effect of biologically-related $c$ on the mean gradients, typically seen on short time scales. However, as demonstrated here, a trade-off is the higher uncertainty. Additional gliders deployed in future missions could potentially decrease the uncertainty associated with $F_{\mathrm{adv}}$ by combination of alternative sampling strategies, leading to more measurements in horizontal space and thus improve the robustness of the plane-fits.

We initially explored using the glider DACs to reference geostrophic shear, required to estimate $F_{\mathrm{adv}}$. However, DAC $v$

velocities were positive throughout most of the deployment period in an area where negative $v$ velocity is expected, as seen in satellite data products (Fig. 8). We think the erroneous DAC $v$ velocities relate to the glider's roll during flight, which was consistently $> 100°$ starboard. The disadvantage of using OSCAR velocities to derive velocity profiles relates to resolution; OSCAR velocities are derived every 8 days on a grid with $0.25°$ resolution.

## 8    Conclusions

This study demonstrates the capability of estimating $N$ using measurements obtained by an autonomous glider. It is the first study of this kind in the Mediterreanean Sea. Both $c(O_2)$ and $c(\mathrm{DIC})$ were used to derive mass budgets, taking into account physical fluxes relating to horizontal advection, air-sea exchange, and mixing.

We showed that advection can be significant in the area of our study and should not be ignored when estimating daily and monthly $N$. Daily mean $N$ estimated from point measurements typical of fixed moorings were significantly affected by

advection on some days. Advection may also be relevant on time-scales longer than one month, but the current study was too short to explore this. However, when the advection flux was averaged over the 25 day-period, there was less of an effect on $N$, suggesting that at least in this region, its importance might vane for longer integration periods.

The spring bloom started in this area of the northwestern Mediterranean Sea around 19 March 2016. This bloom start date was anticipated because phytoplankton blooms occurred in either March or April in the past. However, this was the first time



that high-resolution vertical profiles covering the wider DyFAMed area provided insights into the biogeochemical and physical

processes during a spring bloom.

This second test deployment of prototype ISFET sensors on gliders still showed considerable problems with drift and need

for calibration. Since then, Saba et al. (2018) had more success with an ISFET sensor developed by Sea-Bird Scientific.

However, this sensor is not yet commercially available for gliders.

*Data availability.*

BOUSSOLE buoy data: http://www.obs-vlfr.fr/Boussole/

Meteorological buoy data: Data are archived in the database 'SErvice de DOnnées de l'OMP (SEDOO) Mistrals' (http://mistrals.sedoo.fr/Data-Download)

All glider data will be archived at the British Oceanographic Data Centre (BODC, https://www.bodc.ac.uk/data/bodc_

database/gliders) prior to article acceptance.

**Appendix A: Discrete Water Samples**

Water samples used to measure $c(O_2)$ were collected at the DyFAMed site on 7 March ($n = 10$) and 16 April 2016 ($n = 8$)

using 12 L Niskin bottles (General Oceanics 1010X) from the top 1000 m of the water column; four samples each were from

the top 100 m (Fig. B1b). Reagents needed for the fixation of oxygen were added to the samples at the time of water sample

collection onboard the ship. An automated Winkler titration method with endpoint detection was used after each cruise in the

laboratory at the Observatoire Océanologique de Villefranche sur Mer, France, to determine $c(O_2)$. Replicates were obtained

to determine precision. $c(O_2)$ measured by the rosette-mounted SBE43 sensor within the top 150 m were 15 mmol m$^{-3}$ higher

than the Winkler measurements (Fig. B1b). SBE 43 sensor $c(O_2)$ was corrected by regressing against the Winkler $c(O_2)$

values. The resulting regression coefficients were $0.892 \pm 0.009$ for slope and $(18.7 \pm 2.0)$ mmol m$^{-3}$ for intercept on 7 March

($r^2 = 0.9993$; $\sigma = 0.9$ mmol m$^{-3}$). The corresponding values on 16 April were $0.920 \pm 0.008$ and $(11.8 \pm 1.6)$ mmol m$^{-3}$

($r^2 = 0.9997$; $\sigma = 0.7$ mmol m$^{-3}$).

A Marianda Versatile INstrument for the Determination of Titration Alkalinity (VINDTA 3C; https://www.marianda.com)

was used to measure $c(DIC)$ and $A_T$ at ten depth levels. 19 bottles of certified reference material (CRM) supplied by Scripps

Institution of Oceanography (San Diego, California, USA) were run to calibrate the instrument. Coulometry following standard

operating procedure SOP 2 was used to measure $c(DIC)$ (Dickson et al., 2007; Johnson et al., 1985), and potentiometric

titration following SOP 3b was used to measure $A_T$ (Dickson et al., 2007).

Nutrients at DyFAMed were measured using water collected in the same Niskin bottles as those used for $c(O_2)$, $c(DIC)$, and

$A_T$. Samples were analysed via a standard automated colorimetry system (Seal Analytical continuous flow AutoAnalyser III)

at Observatoire Océanologique de Villefranche-sur-Mer for $NO_x^-$ (nitrate $NO_3^-$ and nitrite $NO_2^-$ combined) (Bendschneider





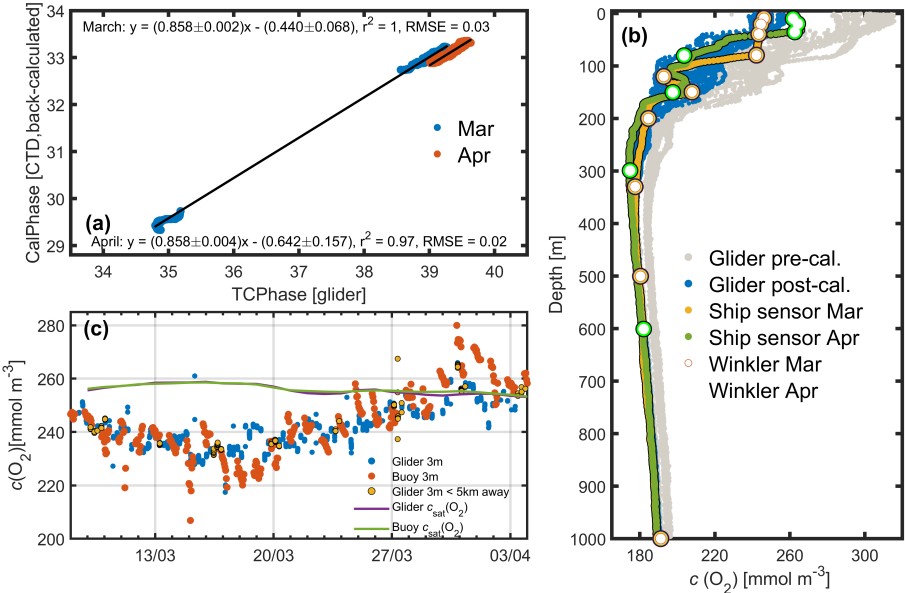

**Figure B1.** Calibration of the glider dissolved oxygen concentration ($c(O_2)$) sensor. (a) The linear regression fit using back-calculated ship pseudo-CalPhase and glider TCPhase for 7 March (blue) and 16 April (orange), along with the corresponding linear equations, $r^2$ and root mean squared errors (RMSE). The 95% confidence bounds are also shown. (b) Glider $c(O_2)$ measured on 8–9 March 8-9 and on 3–4 April before (grey), and after (blue) calibration, the ship sensor $c(O_2)$ on 7 March (yellow) and on 16 April (green), and the $c(O_2)$ Winkler samples on 7 March (white with red border) and on 16 April (white with green border). (c) All glider sensor $c(O_2)$ at 3 m depth (blue spots), < 5 km away from the BOUSSOLE buoy only (orange spots), and glider $c_{sat}$ (purple line) are compared with the BOUSSOLE buoy sensor $c(O_2)$ measurements (orange spots) at 3 m depth, and buoy $c_{sat}$ (green line).

and Robinson, 1952), Si(OH)$_4$ (Murphy and Riley, 1962) and PO$_4{}^{3-}$ (Strickland and Parsons, 1972), with detection limits of 0.01, 0.02, and 0.02 $\mathrm{mmol\,m^{-3}}$, respectively (de Fommervault et al., 2015).

**Appendix B: Glider calibrations**

**B1  O$_2$ Measurements**

Glider $c(O_2)$ were calibrated to take into account the response time ($\tau$) of the sensor, which is dependent on the thickness, and
usage of the sensor foil (McNeil and D'Asaro, 2014), as well as temperature, and to account for the difference between the glider measurements and $c(O_2)$ obtained by the ship (Fig.  B1b) (Binetti et al., 2020; Hemming et al., 2017). To correct $c(O_2)$ measured by the glider sensor, the sensor-related oxygen engineering parameters TCPhase and CalPhase were used. A mean $\tau$ of 8 seconds was applied to correct measurements for the sensor time lag. This mean $\tau$ value was determined from the lowest root mean square difference between descending and ascending TCPhase profiles shifted in time by values of $\tau$ ranging from 0
to 50 seconds. After lag-correction of the glider TCPhase, the relationship between the ship sensor pseudo-CalPhase measured





on 7 March and on 16 April, and the glider TCPhase measured on 8 March, and on 4 April was determined (Fig. B1a). The ship sensor pseudo-CalPhase was calculated from the calibrated ship sensor $c(O_2)$ by inverting the set of equations normally used to obtain $c(O_2)$ from glider optode TCPhase and CalPhase. The slope and offset coefficients obtained from the relationship between the glider TCPhase over the entire deployment time period and ship sensor pseudo-CalPhase on 7 March and 16

April (Fig. B1a) were linearly interpolated over the duration of the deployment. The interpolated coefficients were matched with glider measurements in time, and used to correct all glider CalPhase profiles obtained during the deployment, allowing a calibration of glider $c(O_2)$ (Fig. B1b). After calibration, the glider $c(O_2)$ agreed with buoy $c(O_2)$ (Fig. B1c), as would be expected because the same Winkler samples were used to calibrate ship and buoy oxygen sensors.

## B2   ISFET pH Measurements

The deployment offered a second opportunity to trial two prototype ISFET pH–$p(CO_2)$ sensor pairs, previously tested on an underwater glider in the Sardinian Sea during the REP14–MED experiment (Hemming et al., 2017; Onken et al., 2018). The sensors are custom-built non-commercially by Kiminori Shitashima's group at Tokyo University of Marine Science and Technology, Japan. ISFET sensors measure pH using the interface potential between a reference chlorine electrode (Cl-ISE) and a semiconducting ion sensing transistor (Hemming et al., 2017).

One pH-$p(CO_2)$ sensor pair was stand-alone, meaning measurements were logged and stored by the sensor and retrieved after the deployment. Another sensor pair was integrated into the glider electronics allowing measurements to be sent remotely by satellite in near-realtime. Both the stand-alone and integrated sensors were positioned on the underside of the glider to limit the effect of sunlight on measurements, and backup batteries were provided to supply power in between sampling (Hemming et al., 2017). Unfortunately, the stand-alone sensor ceased operating after less than 3 days around 0700 CET on 10 March

because of an issue with its power supply. For this reason, $pH_T$ obtained by the stand-alone sensor was not used. The sensors were placed in a bucket of locally-collected coastal surface seawater for a period of 10 hours before deployment. Several weeks of pre-deployment conditioning have been recommended (Bresnahan et al., 2014; Takeshita et al., 2014), but this was not possible due to time constraints.

A comparison of sensor measurements with $pH_T$ calculated from discrete $c(DIC)$ and $A_T$ measurements both during the de-

ployment and with archived data from between 1998 and 2013 at the DyFAMed site (http://mistrals.sedoo.fr/Data-Download), indicated problems in accuracy and stability (Fig. B2). The $pH_T$ of the integrated sensor drifted by 0.8 over the course of the 29 day deployment, with a depth-dependent range in $pH_T$ of 1.

$pH_T$ measurements were corrected for drift using the mean difference (offset) between dive 10 $pH_T$ measured between 350 and 950 m and $pH_T$ measured at the same depths during each subsequent dive. This method assumes that $pH_T$ variations over

the 29 day deployment between 350 and 950 m are negligible with respect to the sensor precision. The offsets were added to the full-depth glider $pH_T$ profiles, and the range of $pH_T$ was reduced threefold to approximately 0.3 (Fig. B2).

After correcting $pH_T$ for drift, $pH_T$ was corrected for pressure using a pressure coefficient of $+5.6 \times 10^{-5}$ dbar$^{-1}$. After adding this correction term, the range of $pH_T$ was reduced to approximately 0.2 (Fig. B2).





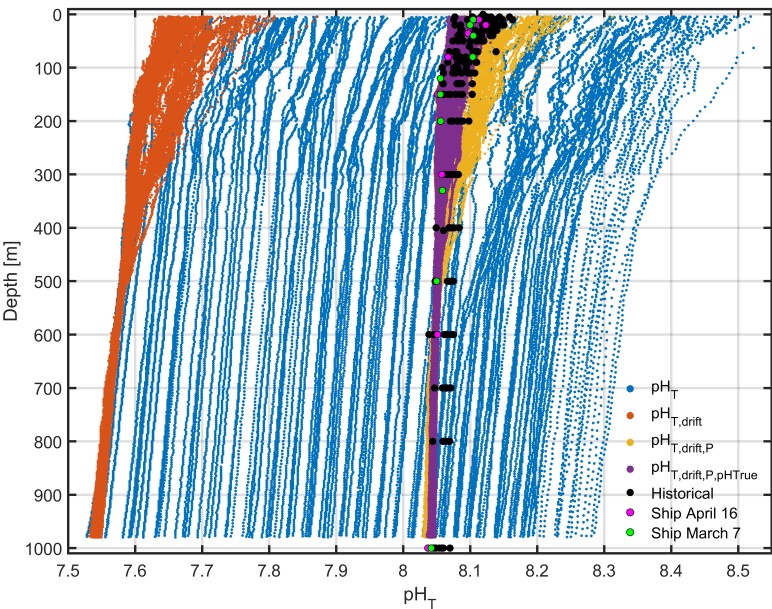

**Figure B2.** A comparison of glider $pH_T$ after various corrections have been applied. Raw $pH_T$ (blue) is corrected for drift (orange), and subsequently for pressure effects (yellow). Finally $pH_T$ is compared with 'true' ship $pH_T$ and corrected. Historical $pH_T$ anomalies (black) collected in March and April between 1998 and 2013 are shown for comparison, alongside the ship $pH_T$ on 7 March and 16 April 2016.

Finally, integrated ISFET $pH_T$ were corrected relative to 'true' discrete $pH_T$. Regression coefficients calculated using glider
drift- and pressure-corrected $pH_T$ and ship $pH_T$ collected on 7 March and on 16 April were used to correct glider $pH_T$. The range of glider $pH_T$ was now matched that of historical discrete samples (Fig. B2).

### Appendix C:  Deriving parameters using CO2SYS

The CO2SYS software package (Van Heuven et al., 2011; Orr et al., 2015) was used to derive discrete $pH_T$ for correcting glider ISFET $pH_T$, using discrete measurements of $c(DIC)$, $A_T$, in situ temperature, salinity, pressure, and $Si(OH)_4$ and $PO_4^{3-}$
concentrations. Equilibrium constants (Mehrbach et al., 1973; Dickson and Millero, 1987), sulfate acid dissociation constant (Dickson, 1990) and total borate concentration (Uppström, 1974) were used, as recommended by previous Mediterranean-based studies (Álvarez et al., 2014; Key et al., 2010). $c(DIC)$ and $A_T$ had an uncertainty of 3.5 $\mu mol\,kg^{-1}$ and 3.2 $\mu mol\,kg^{-1}$, respectively, determined from CRM measurements, representing a combined uncertainty of 0.008 in derived $pH_T$.

The $A_T - S$ relationship was determined using $A_T$ and salinity measurements from discrete ship samples collected in
spring 2016 above the salinity maximum ($n = 20$). This parameterisation, $A_T/\mu mol\,kg^{-1} = (80.72 \pm 10)\,(S - 38) + 2551 \pm 4$ ($r^2 = 0.88$; $\sigma = 5.7\ \mu mol\,kg^{-1}$, was used to obtain $A_T$ from glider and buoy salinity.





Calibrated glider $pH_T$, and $A_T$ were used to derive glider $f(CO_2)$ and $c(DIC)$ using CO2SYS. The mean uncertainties of derived $f(CO_2)$ and $c(DIC)$ were 5.3 µatm and 4.1 µmol kg$^{-1}$, respectively, calculated as the mean of the absolute differences of $f(CO_2)$, and $c(DIC)$ derived using single values of $A_T$ and pH (including the margin of error associated with the $A_T - S$ relationship, and the $pH_T$ pressure correction, respectively).

To obtain buoy $c(DIC)$, observed $f(CO_2)$ at 10 m was used with $A_T$ calculated from $S$. The uncertainty of derived buoy $c(DIC)$ was 6.6 µmol kg$^{-1}$, calculated as the absolute difference between the ship $c(DIC)$ sample collected on 7 March at 10 m depth and average $c(DIC)$ measured by the buoy on the same day.

*Author contributions.* This work resulted from MPH's PhD project at the University of East Anglia under the supervision of JK, KJH, DB and JB. GAL, MCG and KS implemented the ISFET sensors on the glider, and deployed it. JK led the UEA piloting team. DA acquired funding for the Bio-optics and Carbon Experiment (BIOCAREX) that supports the CO$_2$ measurements at BOUSSOLE, and led ship operations and ancillary data collection. The data were analysed by MPH and JK, who also wrote the manuscript. All other authors contributed to further editing and writing.

*Competing interests.* The authors declare that they have no competing interests.

*Acknowledgements.* We thank V. Vellucci, E. Diamond, and M. Golbol for monthly sampling at the BOUSSOLE mooring, and for aid in planning the 2016 glider deployment. Michael Hemming's PhD project was funded by the UK Defence Science and Technology Laboratory (DSTL), in cooperation with Direction Générale de l'Armement (DGA, France), with oversight provided by Tim Clarke and Carole Nahum. This work was partially funded by the UK National Environment Research Council (NERC; grant numbers NE/K002473/1, NE/N018699/1). BOUSSOLE is funded by the European Space Agency (ESA / ESRIN contracts 4000102992/11/I-NB and 4000111801/14/I-NB), the Centre National d'Etudes Spatiales (CNES), and this particular study through a grant from the Agence Nationale de la Recherche (ANR, Paris) to the Bio-optics and Carbon Experiment (BIOCAREX). Ship time on R/V Téthys II is provided by the Institut National des Sciences de l'Univers (INSU).



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
