# Peer review of "Net community production in the northwestern Mediterranean Sea from glider and buoy measurements"

_Ocean Science, 2022_

## Author Response (AR1)

**Response to reviewer #1**

The presentation quality was very high. There was a lot of data to communicate in the figures. With a bit of study, they proved clear and informative. The results and conclusions were very clear, if the discussion a little short.

We are pleased that the reviewer found the manuscript interesting to read and we would like to thank them for the helpful comments. Please see below our comments on the discussion.

Please see below details of our changes to the manuscript and refer to the PDF with tracked changes. Line numbers below correspond with the revised manuscript PDF.

specific comments

There were some interesting details on correcting glider data, correcting the pH sensor for drift and validation of sensors with bottle data. There could have been a clearer description at the start of the paper on the motivation to calculate NCP in two ways and on the significance of the discrepancies found. However, the paper successfully argued for augmenting long time series data with targeted short duration profilers and there are only minor changes required.

In an effort to be clearer, we will add the following on line 9:

" …. using two similar methodologies."

and on line 18:

" … but also demonstrates the difficulty in estimating N, …"

We have added these words on line 9 and 18.

technical corrections

L.11 slight change in wording: 'calculation of advective'

We have changed this to:

"… allowed the calculation of advective $O_2$ and DIC fluxes."

We have modified line 11 as above.

L.24: Reference required for processes and climate change?

We will include the following reference:

Bauer, James E., et al. "The changing carbon cycle of the coastal ocean." Nature 504.7478 (2013): 61-70.

This reference is now included in the manuscript on L26.

L.30: Reference for Revelle factor

We will add the following reference on this line:

Broecker, Wallace S., et al. "Fate of fossil fuel carbon dioxide and the global carbon budget." Science 206.4417 (1979): 409-418.

This reference is now included in the manuscript on L31.

L.45: reference on significance of POC and DOC changes? (explain acronyms here)

We will change the text to as follows:

> "Significant increases of Particulate and Dissolved Organic Carbon (POC and DOC) concentrations have been observed during bloom events (e.g. Carlson et al. (1998))."

We have changed the text on L46.

L.108: not sure what B2,C refers to here?

We will update the text to reference Appendix B2.

We have updated the text on L109.

L.173: Interesting observation on the differences in PP and NCP were not explored?

The focus of this study is to estimate NCP using glider observations over a spatial domain. Chlorophyll *a* concentrations can be both an indicator of biomass (accumulated production) and productivity (instantaneous rates), and we use them in a qualitative manner here as indicator of likely patterns in primary productivity, and not to quantitatively compare PP with NCP. We suspect that any differences may be related to differences in phasing, temporal and spatial aggregation scales, but there are too many unknowns involved to draw firm conclusions. Nevertheless, like the reviewer, we consider it to be an interesting observation.

No changes have been made.

L.111: change yer to year

We will correct this in the revised version.

We have corrected this on L112.

L.123: Use acronym here?

We will use 'POC' here.

We now use the acronym on L124.

L.268: the non-Redfield ratios were interesting but not really discussed further (the discussion section is in fact quite small)

There were a few words missing on l. 262 and 264. We will insert "for phosphorus" after "0.4 to 0.7 mmol m$^{-2}$ d$^{-1}$" (l. 262) and "(based on phosphorus)" after "72 mmol m$^{-2}$ d$^{-1}$" (l. 264).

We found that the last two sentences on l. 266 and 267, partially contradicted the previous one, i.e., as far as the nutrient drawdown is concerned, the N(DIC) values are in reasonable agreement with the N- and P-based values (within the uncertainties and considering the omission of advective nutrient fluxes). We will therefore rephrase this as:

"The values are in reasonable agreement with the $N$(DIC) values derived for the bloom period between 19 March and 3 April of (85±98) to (128±90) mmol m$^{-2}$ d$^{-1}$), indicating nutrient consumption in line with the assumed stoichiometric ratio." and delete the sentence beginning "However, the N-based value ...".

The subsequent sentence will be moved to the next paragraph and the word "even" removed, i.e.

"The discrepancy between expected and observed stoichiometry is larger for the O$_2$-based N values, especially for the glider-based observations."

The subsequent discussion is as exhaustive as we feel we can justify from our current understanding. However, we will add another citation to note that other studies have also found deviations from Redfield ratios for nitrate and oxygen-based NCP (Hull et al. (2021)

Hull, Tom, et al. "Simultaneous assessment of oxygen-and nitrate-based net community production in a temperate shelf sea from a single ocean glider." *Biogeosciences* 18.23 (2021): 6167-6180.

We have made changes to the text on L263-269 as described above, and we have added a sentence relating to the study by Hull et al (2021) on L270.

L.295: is this change in concentration (just written as 'c')?

Yes. For clarity we will change this to 'concentrations'.

We now write 'concentration/s' on L294 and L299.

L.312: what do you mean by vane?

We meant 'wane', but we will now use 'diminish' instead.

We now use 'diminish' on L334.

L.401: close the brackets

We will do this in the revised version.

We have closed the brackets on L429.

**Response to reviewer #2**

This manuscript presents a detailed account of glider and buoy measurements of dissolved $O_2$ and inorganic carbon (DIC) concentrations, which, together with physical fluxes due to horizontal advection, air-sea exchange, and mixing, are used to calculate net community production (N) in the euphotic zone of the NW Mediterranean Sea in spring 2016. Both the data, which are or will be openly accessible through BOUSSOLE, MISTRALS and BODC databases, and the derived N rates are important contributions to our knowledge on the biotic contribution to the ocean carbon sink, which is an important and timely scientific topic. The presentation of methods is detailed and comprehensive, including calibration of sensor, and results and calculations are also clearly presented. I am however less enthusiastic about the conclusions and discussion.

We are pleased that the reviewer found the manuscript interesting to read and we would like to thank them for the helpful comments. Please see below our comments on the discussion.

Please see below details of our changes to the manuscript and refer to the PDF with tracked changes. Line numbers below correspond with the revised manuscript PDF.

Conclusions seem a little vague to me, mainly focus on what has been done, rather than what has been observed. E.g., "this was the first time that high-resolution vertical profiles covering the wider DyFAMed area provided insights into the biogeochemical and physical processes during a spring bloom." However which are those insights provided by this study is not mentioned.

We will reorganise the conclusion section and add some further details.

We have reorganised the conclusion section and added some text. Please refer to the manuscript PDF with tracked changes.

I particularly don't agree with the conclusion that "this study demonstrates the capability of estimating N using measurements obtained by an autonomous glider". I do fully support the approach and agree that gliders have a unique potential to provide estimates of N at spatial and temporal scales that are unsuitable for other methods/platforms.

We will modify the text on line 305 as follows:

> "This study demonstrates the **potential** of using autonomous glider measurements for estimating N."

We have modified the text on L316.

In this regard, I think that the paper is a very substantial scientific contribution. I also acknowledge the conscientious calculations of both rates and uncertainty, and their detailed description in the paper.

Thank you.

[…]I think that the paper would improve with a discussion on the causes for the large differences between the different N estimates, which are the rates that better explain the backscatter and chlorophyll-a build-up in the region, and a more complete comparison of bloom N estimates with the literature (including blooms elsewhere). Results here are only compared with Coppola et al. (2018)

and Copin-Montégut (2000), yet it is acknowledged that such a comparison is not possible "because each study is focused on different timescales (from years to days) or different seasons."

The above comments provided by the reviewer highlight that calculating $N$ and advection using glider pH and $O_2$ measurements is possible, but difficult. Bearing in mind their uncertainties the results are mutually compatible. Differences primarily arise from challenges relating to glider spatial coverage and sensor calibration. We have not shied away from this, considering that we have discussed such issues and potential contradictions in the manuscript. For example, we discuss the uncertainty of $N$ in detail on lines 289 – 298. Previous studies may also have been too optimistic about uncertainties associated with their estimates.

We'd also like to refer to our discussion on l. 283, which refers to the good agreement between the March and April data of Coppola et al. (2018) and our results. The notion that there are large differences between $N$ estimates is perhaps overstated; we would rather conclude that there are large uncertainties in $N$ estimates, and that deriving $N$ estimates from in situ measurements with a comprehensive uncertainty budget is comparatively hard.

Relating to the length of the discussion, it is also worth noting that some discussion has appeared outside of Section 7. For example, on lines 169 – 173 we discuss the opposing north-south gradients for glider O2 and satellite-derived ocean color, and on lines 264 – 271 we discuss the non-Redfield ratios in the context of previous studies and Redfield-derived carbon fluxes.

It is challenging to determine which $N$ estimate is "better", however we will add a short paragraph further discussing the potential sources of errors for the $N$ estimates. We will also add a conclusion that it is important to consider all systematic and random uncertainties when deriving $N$ estimates.

We have added a short paragraph starting on L308 and a sentence on L320 in the Conclusion section regarding $N$ uncertainties.

We have also included some discussion relating to phytoplankton blooms elsewhere estimated using glider measurements. We have added a new paragraph to the discussion starting on Ln293.

- Figure 1 presents surface chlorophyll a concentrations on 24 March 2016. I think that the climatology for the period of study or the bloom period would be more useful as context for N estimates.

We show chlorophyll a concentrations on 24 March 2016 to highlight the patchiness of blooms in this region. This was useful for the discussion on lines 169 – 173 where we reference the figure.

- Which are the consecuentes of using a single mean euphotic depth of 46 m for calculating N throughout the study. Large temporal differences in backscatter (Fig.5) and spatial differences in chlorophyll-a (Fig.1) suggest that the actual euphotic depth should have changed substantially during the period of study, particularly associated to the phytoplankton bloom. Do this have an impact on N estimation under different scenarios? More specifically, is the calculated N an unbiased estimation of euphotic zone net community production both during periods when ZeuZlim? On the other hand, photosynthetic gross production (GP) is limited to the euphotic layer, however the respiration (R) of the organic matter produced is not; beyond

entrainment, do the large changes observed in the ratio between the euphotic and mixed depths (Fig.5) have an effect in the interpretation of N?

We have investigated the sensitivity of $N$ estimates to $z_{lim}$ (See figure 1 below). The overall pattern is similar in each case, however the choice of $Z_{lim}$ can at times affect $N$, particularly for DIC-based $N$ estimates (e.g., after 25/03). The largest absolute difference between single-day $N$ estimates using a $z_{lim}$ of 36 m and 56 m was 135 mmol m$^{-2}$ d$^{-1}$ on 26/03 for $N_b(DIC)$.

[Figure]

**Figure 1** Comparing $N_g$ and $N_b$ estimates using different $Z_{lim}$.

We will add a brief discussion on this to the revised manuscript.

The 4-day smoothed MLD is $> Z_{eu}$ most of the time before 25/03, and $< Z_{eu}$ afterwards. We have found no relationship between the MLD/ $Z_{eu}$ ratio and the $N$ sensitivity to $z_{lim}$.

We have added a sentence on L313 relating to the above result.

---

## Referee Report (RR1)

General Comments

The manuscript uses glider and buoy measurements of dissolved O2 and inorganic carbon (DIC) concentrations along with horizontal advection, air-sea exchange and mixing, to compute net community production (N) in the euphotic zone of the NW Mediterranean Sea in spring 2016. I think it is a good manuscript and very intriguing.

The work to assess/ compute the N rates provides us with an important building block for a better understanding on the biotic contribution to the ocean carbon sink, which is a hot topic. The presentation of methods is well organized and explained in detail the work done.

After the revision following the two reviewers, I still agree with the fact that the conclusions do not give proper importance to the work done both in terms of the method and the conclusions that come out of the work. They turn out not to be incisive as they do not summarize in broad terms the results achieved.

I think the aspects that still constitute the uncertainties should also be explained in an organized way.

Specific comments

I noticed that the reviewers' suggestions were followed.

I would have 2 more suggestions: in the caption of figure 1 I would change 'the position of each glider point' to 'glider track' since you can't see the surfacing.

line 172 add Fig. to 1b

---

## Author Response (AR2)

General Comments

*The manuscript uses glider and buoy measurements of dissolved O2 and inorganic carbon (DIC) concentrations along with horizontal advection, air-sea exchange and mixing, to compute net community production (N) in the euphotic zone of the NW Mediterranean Sea in spring 2016. I think it is a good manuscript and very intriguing.*

*The work to assess/ compute the N rates provides us with an important building block for a better understanding on the biotic contribution to the ocean carbon sink, which is a hot topic. The presentation of methods is well organized and explained in detail the work done.*

*After the revision following the two reviewers, I still agree with the fact that the conclusions do not give proper importance to the work done both in terms of the method and the conclusions that come out of the work. They turn out not to be incisive as they do not summarize in broad terms the results achieved. I think the aspects that still constitute the uncertainties should also be explained in an organized way.*

We have revisited the conclusion section, adding extra text (e.g. on uncertainties), reorganising the paragraphs and sentences, and rewriting some text. We think that the conclusion section now summarizes our results nicely and is more neat.

Please see the tracked version of the manuscript to see our changes.

Specific comments

*I noticed that the reviewers' suggestions were followed.*

*I would have 2 more suggestions: in the caption of figure 1 I would change 'the position of each glider point' to 'glider track' since you can't see the surfacing.*

Done.

*line 172 add Fig. to 1b*

Done.